# Quality Evaluation of Lonicerae Flos Produced in Southwest China Based on HPLC Analysis and Antioxidant Activity

**DOI:** 10.3390/molecules29112560

**Published:** 2024-05-29

**Authors:** Qundong Liu, Huanhuan Yu, Yuzhuo Dong, Wenjing Quan, Zhimin Su, Longyun Li

**Affiliations:** 1Chongqing Academy of Chinese Materia Medica, Chongqing 400065, China; lettuceliu@outlook.com (Q.L.);; 2Chongqing Institute for Food and Drug Control, Chongqing 401121, China; 3Research Centre of Natural Resources of Chinese Medicinal Materials and Ethnic Medicine, Jiangxi University of Chinese Medicine, Nanchang 330004, China

**Keywords:** *Lonicera macranthoides*, Lonicerae Flos, quality evaluation, chlorogenic acid, saponin, antioxidative activity, multivariate analysis

## Abstract

*Lonicera macranthoides*, the main source of traditional Chinese medicine Lonicerae Flos, is extensively cultivated in Southwest China. However, the quality of *L. macranthoides* produced in this region significantly varies due to its wide distribution and various cultivation breeds. Herein, 50 Lonicerae Flos samples derived from different breeds of *L. macranthoides* cultivated in Southwest China were collected for quality evaluation. Six organic acids and three saponin compounds were quantitatively analyzed using HPLC. Furthermore, the antioxidant activity of a portion of samples was conducted with 2,2′-Azinobis-(3-ethylbenzthiazoline-6-sulphonate) (ABTS) and 1,1-diphenyl-2-picryl-hydrazyl (DPPH) radical scavenging experiments. According to the quantitative results, all samples met the quality standards outlined in the Chinese Pharmacopoeia. The samples from Guizhou, whether derived from unopened or open wild-type breeds, exhibited high quality, while the wild-type samples showed relatively significant fluctuation in quality. The samples from Chongqing and Hunan demonstrated similar quality, whereas those from Sichuan exhibited relatively lower quality. These samples demonstrated significant abilities in clearing ABTS and DPPH radicals. The relationship between HPLC chromatograms and antioxidant activity, as elucidated by multivariate analysis, indicated that chlorogenic acid, isochlorogenic acid A, isochlorogenic acid B, and isochlorogenic acid C are active components and can serve as Q-markers for quality evaluation.

## 1. Introduction

Lonicerae Flos, or “shanyinhua” in Chinese, has a long history of use in traditional Chinese medicine for treating fever and respiratory infections [1]. In addition, Lonicerae Flos exhibits diverse pharmacological properties, including anti-inflammatory, antimicrobial, antiviral, and anticancer effects [2,3,4]. Recently, it has been mainly used to prevent and treat respiratory syndromes and viral infections, such as HINI influenza, hand-foot- and-mouth disease, and COVID-19 [5]. According to the Chinese Pharmacopoeia, Lonicerae Flos refers to the dried flower and flower bud of *Lonicera macranthoides* Hand.-Mazz., *L. hypoglauca* Maq., *L. confusa* DC., or *L. fulvotomentosa* Hsu et S. C. Cheng containing no less than 2.0% chlorogenic acid (CGA) and 5.0% of the total contents of macranthoidin B (MB) and dipsacoside B (DB) [6]. Among these botanical sources, *L. macranthoides* has emerged as the main source of Lonicerae Flos in the current market owing to its extensive distribution and cultivation in southern China [7].

The genus *Lonicera* (Caprifoliaceae) comprises approximately 180 species distributed throughout the Northern Hemisphere. Multiple compounds belonging to the flavonoid, terpenoid, and iridoid groups were detected in *Lonicera* species; however, most of the species showed high diversity in genetics and chemical compositions [8,9]. Lonicerae Flos, derived from *L. macranthoides*, is rich in phenolic acids, flavonoids, saponins, and essential oils. Its medicinal properties primarily stem from phenolic acids and saponins [10]. CGAs, the main phenolic acids in *L. macranthoides*, are ester compounds composed of various *trans*-cinnamic acids [11]. Various CGAs have been identified in several plant species, particularly edible plants, such as coffee [12], black tea [13], roses [14], chrysanthemum [14], potatoes [15], burdock [16], and some other fruits or vegetables [17]. Owing to their protective effects, CGAs play a crucial role in plant resistance to environmental stressors, such as ultraviolet light [18], insect pests [19], and fungal infections [20]. As the main phenolic acids in *L. macranthoides*, CGAs exhibit various biological activities, including antimicrobial [21], anti-inflammatory [22], antioxidant [23], and antiobesity effects [24]. Saponins, another important group of compounds in *L. macranthoides*, are classified into six major types based on their chemical structures: hederin-type, oleanane-type, ursane-type, lupane-type, fernane-1-type, and fernane-2-type [25]. These saponins display diverse biological activities, including anti-inflammatory, antibacterial, antiallergic, immunomodulatory, and antitumor properties, as well as protection against Alzheimer’s disease [26,27]. Medicinal plants are regarded as a readily accessible and powerful source of antioxidants due to their diverse chemical compounds, which can operate independently or synergistically to treat diseases and enhance health. [28]. The antioxidant activity of *L. macranthoides* is attributed to its ability to scavenge free radicals, which are involved in developing various diseases [29]. This suggests that the antioxidant activity of *L. macranthoides* may underlie its medicinal properties [30].

*L. macranthoides* has traditionally been cultivated in Southwest China, spanning regions such as Chongqing City, Sichuan Province, Hunan Province, and Guizhou Province (Figure 1a). Consequently, certain breeds of *L. macranthoides* have been observed and cultivated in these regions, becoming the predominant breeds. Owing to variations in climate, geography, and cultivation methods in these regions, most of these breeds exhibit distinct morphological characteristics in their flowers. For example, some breeds retain a bud-like appearance without splitting, or rarely split, maintaining their color until they wither (Figure 1b) [31]. These unopened mutant breeds of *L. macranthoides* are now widely cultivated. During our field investigation, the open wild-type breeds are still the main cultivation breeds in Guizhou, characterized by flower buds that split and change from white to yellow as they mature (Figure 1c). The quality of the flowers from open wild-type breeds varies during their development stages, with higher CGA content observed in the early stages than in the late stages [32]. Considering these factors, the quality distinctions arising from the diverse producing regions and breeds of *L. macranthoides* remain unclear, and the variations in components may impact its antioxidant activity. Therefore, developing a comprehensive quality evaluation method that compares the components and antioxidant activity of *L. macranthoides* from different producing regions and breeds is necessary.

Herein, with an aim to clarify the quality differences of Lonicerae Flos derived from *L. macranthoides* cultivated in Southwest China, 50 samples were collected for investigation. CGA and other phenolic acid compounds were quantified using HPLC analysis (Figure 2). Three saponin compounds in *L. macranthoides*—MB, macranthoidin A (MA), and DB—were quantitatively determined using HPLC coupled with an evaporative light-scattering detector (ELSD) owing to low ultraviolet absorption. Component variation was characterized using multivariate statistical analysis, specifically orthogonal partial least squares-discriminant analysis (OPLS-DA). Furthermore, 2,2′-Azinobis-(3-ethylbenzthiazoline-6-sulphonate) (ABTS) and 1,1-diphenyl-2-picryl-hydrazyl (DPPH) scavenging assays were conducted to evaluate the antioxidant activity of the cultivated samples. Principal component analysis (PCA), partial least squares (PLS) regression, and heatmap analysis were conducted to identify active components and identify potential correlations between the HPLC chromatograms and antioxidant activity.

## 2. Results and Discussion

### 2.1. Method Validation

The quantitative method was validated by evaluating linearity (r), precision, stability, repeatability, and recovery (Table 1). High correlation coefficient values were observed for all nine analytes (r ≥ 0.9993 for the six phenolic acid compounds and r ≥ 0.9926 for the three saponin compounds), indicating strong correlations between concentrations and peak areas within the tested ranges (from 0.002 to 1.903 mg/mL). Precision, repeatability, and stability were also validated for each analyte, with RSD varying from 0.79% to 1.95%, 0.13% to 2.60%, and 0.34% to 1.66%, respectively. The recoveries of the nine analytes ranged from 93.99% to 100.67%, with RSD varying from 1.13% to 1.91%. These findings demonstrate that the established method is sensitive and sufficiently accurate for quantitatively analyzing the nine compounds in *L. macranthoides*.

### 2.2. Metabolite Profiling of L. macranthoides

Six phenolic acid compounds including CGA, neochlorogenic acid (NA), cryptochlorogenic acid (CA), isochlorogenic acid A (IAA), isochlorogenic acid B (IAB), isochlorogenic acid C (IAC) and three saponin compounds (MB, DB, and MA) were simultaneously detected in the HPLC chromatograms of all samples derived from *L. macranthoides* by comparing retention times. A representative HPLC chromatogram is shown in Figure 3. No variations in component composition were observed among the *L. macranthoides* samples derived from different producing regions of China, indicating consistency in the composition of the samples used in this study.

### 2.3. Quantitative Analysis of the Nine Compounds in L. macranthoides

The contents of the nine compounds detected in the HPLC chromatograms were quantified using linear formulas for each compound. As presented in Table 2 and Figure 4, CGA, NA, CA, IAA, IAB, IAC, MB, DB, and MA exhibited relative contents ranging from 3.296% to 7.839%, 0.450% to 1.144%, 0.117% to 0.490%, 1.671% to 4.582%, 0.056% to 0.264%, 0.061% to 0.139%, 3.551% to 13.652%, 0.836% to 1.789%, and 0.350% to 1.202%, respectively. In addition, the total relative contents of MB and DB across all samples ranged from 4.989% to 15.000% (Table 2). According to the Chinese Pharmacopoeia (2020), Lonicerae Flos should contain no less than 2.0% CGA and 5.0% of the total contents of MB and DB, calculated with reference to the dried drug [6]. Our results demonstrate that all samples used in this study met the quality standard. Only Sample SYH5 from Suiyang, Guizhou, exhibited a slightly lower content of two saponins (4.989%) than the quality standard. XIAO et al. [33] detected CGA in 50 Lonicerae Flos derived from *L. macranthoides* cultivated in Longhui, Hunan, using HPLC, with the relative content ranging from 3.07% to 7.09%. Wu et al. [34] collected flowers of *L. macranthoides* from three regions in Hunan and Xiushan, Chongqing, and found that CGA showed a relative content of 3.067–4.479%. Liu et al. [35] collected *L. macranthoides* samples from Hunan, Sichuan, and Hubei and found that these samples contained CGAs of 72.14, 43.91, and 28.87 mg/g, respectively. These results align with our findings. Sun et al. [36] established a method for simultaneously determining 5 saponins in 10 *L. macranthoides* samples produced in Longhui, Hunan. MB, MA, and DB exhibited relative contents ranging from 4.317% to 6.185%, 0.045% to 0.094%, and 0.6245% to 1.342%, respectively, similar to our results.

The total contents of the six phenolic acids or three saponin compounds found in samples cultivated in different regions significantly differed (Table 2). When summing the contents of the six phenolic acids, samples from Chongqing City exclude Xiushan County (CQ1-11); Xiushan County, Chongqing City (XS1-10); and Longhui County, Hunan Province (LH1-6) showed average contents of 7.502%, 8.073%, and 8.908%, respectively, with no significant difference. Samples from Bazhong City, Sichuan Province (BZ1-3) showed an average content of 6.478%, significantly lower than those produced in Chongqing or Hunan. Samples from unopened type in Suiyang County, Guizhou Province (SY1-2) and wild open type in Suiyang County, Guizhou Province (SYH1-18) showed average contents of 12.059% and 10.009%, respectively, which is significantly higher than the other samples, with SY1-2 (12.059%) exhibiting a significantly higher content than SYH1-18 (10.009%). Regarding the total contents of the three saponin compounds, samples CQ1-11, XS1-10, LH1-6, BZ1-3, SY1-2, and SYH1-18 showed similar average contents of 8.485%, 8.288%, 8.367%, 8.901%, 9.993%, and 8.795%, respectively. Tian et al. [37] suggested that environmental factors, such as sunlight, temperature, water, altitude, and soil nutrition, may impact the content of CGA in tobacco. Zhang et al. [38] analyzed climatic factors affecting CGA in Lonicerae Japonica Flos and found that high-temperature stress had the most significant influence. Chen et al. examined the relationship between Lonicerae Flos quality and altitude. Their results showed that altitude significantly influenced the quality of Lonicerae Flos derived from *L. macranthoides*, with pharmacological substances increasing as altitude increased. Altitude also influenced the contents of MB and DB, with MB showing the highest content in samples from 600 to 700 m and DB showing the highest content in samples from 800 to 900 m [39]. To our knowledge, the altitudes of Chongqing, Bazhong, and Longhui range from 200 to 400 m, whereas Suiyang has an altitude of approximately 1300 m, which is much higher than the other regions. This disparity may be the main factor related to the high CGA content in *L. macranthoides* cultivated in Suiyang.

The HPLC results revealed that *L. macranthoides* cultivated in Southwest China exhibited high quality owing to their elevated levels of phenolic acids and saponin compounds. Samples from Chongqing, Hunan, and Sichuan demonstrated consistent contents of all nine compounds, indicating uniform quality. However, samples from Guizhou exhibited relatively high quality, with the total contents of the six phenolic acids and three saponin compounds ranging from 7.464 to 10.009% and 8.795 to 16.015%, respectively. Sample SYH18, which had the highest content of saponin compound, and Sample SYH4, which had the lowest content, belonged to the SYH group. This indicates significant fluctuations in quality for Lonicerae Flos derived from open wild-type *L. macranthoides* in Guizhou, possibly owing to genetic diversity.

OPLS-DA was conducted on the HPLC data of all 50 *L. macranthoides* samples. The score and loading plots of OPLS-DA are depicted in Figure 5. In the score plot, samples produced in Suiyang, Guizhou, including those derived from the unopened and open wild types, were distinguished from the others. Among the remaining samples, three from Bazhong, Sichuan were separated from those produced in Chongqing and Longhui, Hunan. Samples from Chongqing and Longhui exhibited partial overlap. Samples from the entire Chongqing region, including those from Xiushan and other regions, were indistinguishable, indicating a similar quality. The results of the OPLS-DA aligned with the HPLC analysis.

### 2.4. Antioxidant Activity

All samples exhibited high radical scavenging activity in the ABTS and DPPH assays, with radical clearing ratios ranging from 68 to 93% in the ABTS assay and 48 to 71% in the DPPH assay (Table 3). For each sample group, samples within CQ, XS, LH, BZ, and SY demonstrated average ABTS radical scavenging as activities of 76%, 82%, 84%, 71%, and 92%, respectively, and average DPPH radical scavenging activities of 57%, 64%, 61%, 54%, and 71%, respectively. These results align with the tendencies observed in the bioactive compound contents. XIAO et al. [33] assessed radical scavenging activity based on DPPH for 50 Lonicerae Flos samples derived from *L. macranthoides* cultivated in Longhui, Hunan, using HPLC. Their results showed clearing ratios ranging from 21.87 to 86.74%, which are consistent with our results (48–71%).

### 2.5. Relationship between HPLC Chromatograms and Antioxidant Components

PCA, heatmap analysis, and PLS regression were conducted to explore the relationship between HPLC chromatograms and antioxidant activity. These statistical analysis models used the same data matrix created from HPLC peak areas and ABTS/DPPH scavenging activity data of 32 *L. macranthoides* samples. The PCA, heatmap analysis, and PLS regression results are depicted in Figure 6, Figure 7 and Figure 8, respectively.

The PCA results are presented in Figure 6, where the crude drug samples are divided into five groups based on their producing regions. This classification aligns with the findings shown in Figure 5a. Notably, the variables of ABTS/DPPH scavenging activity and a portion of compounds (CGA, IAA, IAB, and IAC) tend to cluster, observed above and below the x-axis in quadrants 1 and 4, indicating that samples produced in Suiyang exhibit the strongest antioxidant activity and the highest CGA, IAA, IAB, and IAC contents. This observation is consistent with the results of the antioxidant activity and quantitative analyses, indicating a potential relationship between antioxidant activity and these four compounds.

To further elucidate the antioxidant active components in *L. macranthoides*, a heatmap using Pearson’s correlation coefficient was used to explore the correlation between antioxidant activity and the compounds. As shown in Figure 7, ABTS scavenging activity exhibited relatively strong correlations with CGA (r = 0.73), IAA (r = 0.73), IAB (r = 0.77), and IAC (r = 0.70). Similarly, DPPH scavenging activity showed relatively strong positive correlations with CGA (r = 0.71) and IAB (r = 0.66) while displaying positive correlations with IAA (r = 0.56) and IAC (r = 0.48). These findings indicate that CGA, IAA, IAB, and IAC are associated with ABTS and DPPH radical scavenging activity.

PLS regression was also used to investigate the correlation between antioxidant activity and the compounds. The compounds served as X variables, whereas either ABTS or DPPH radical scavenging activity served as Y variables to construct the PLS model (Y variables). Variable influence on projection (VIP) values are often used to assess the importance of the X variables in explaining the Y variables, with values >1 considered more relevant [40,41]. In Figure 8, the four red columns representing CGA, IAA, IAB, and IAC can be observed on the left side of the graph owing to their VIP values being >1, indicating a strong correlation between these four compounds and antioxidant activity. The regression coefficients in the PLS model reflect the degree of contribution of X to Y variables [42], which was also used to identify the compounds with a high contribution to the PLS model. In Figure 8, CGA, IAA, IAB, and IAC exhibited a positive relation and a high contribution to the model, suggesting that these four compounds might be the active components in *L. macranthoides*. These results were consistent with those of the heatmap in Figure 7.

Sato et al. [43] reported on the antioxidant activity of CGA, confirming the reliability of our results. Additionally, IAA and IAC showed DPPH scavenging activity similar to that of IAB at equivalent concentration levels [44,45]. Therefore, the moderate positive correlations for IAA and IAC might arise from differences in the concentrations of these three compounds or alterations in their synergistic effects owing to varying concentration ratios.

The above results reveal that CGA, IAA, IAB, and IAC are antioxidant-active compounds in *L. macranthoides*, and these four compounds could serve as indicators for quality evaluation. Based on these findings, active components can be rapidly identified by characterizing the correlation between characteristic components and antioxidant activity. Our results also suggest that combining multiple chemometric methods is fast and effective for exploring active components, identifying Q-markers, and elucidating the relationship between fingerprints and active components.

## 3. Materials and Methods

### 3.1. Materials

The 50 Lonicerae Flos samples derived from *L. marcanthoides* used in this study were collected in 2022 (Table 4). Among these, 21 samples were sourced from Chongqing, China, with 10 samples originating from Xiushan County, Chongqing, China, and 11 from other regions within Chongqing, China. In addition, six samples were collected from Longhui County in Hunan Province, China, and three from Bazhong City in Sichuan Province, China. All aforementioned samples were derived from unopened breeds of *L. macranthoides*. Furthermore, 2 samples were sourced from unopened breeds, whereas 18 samples were sourced from open wild-type breeds (primarily in the early stages) of *L. macranthoides* in Suiyang County, Guizhou Province, China. All samples were sealed in plastic bags at room temperature (15 to 25 °C) to avoid sunlight exposure. Vouchers were deposited at the Chongqing Academy of Chinese Materia Medica in Chongqing, China.

### 3.2. Chemicals and Reagents

Chemical standards of CGA, NA, CA, IAA, IAB, IAC, MB, MA, and DB were purchased from Shanghai Yuanye Biotechnology Co., Ltd. (Shanghai, China) with purities of ≥98%.

HPLC-grade methanol and acetonitrile were purchased from TEDIA (Fairfield, OH, USA), whereas HPLC-grade formic acid was procured from Aladdin (Shanghai, China). Analytical-grade methanol was sourced from Chongqing Chuandong Chemical (Group) Co., Ltd. (Chongqing, China). In addition, K_2_S_2_O_8_ was purchased from Shanghai Macklin Biochemical Co., Ltd. (Shanghai, China), whereas ABTS and DPPH were obtained from Shanghai Yuanye Biotechnology Co., Ltd. (Shanghai, China). Ultrapure water (18.2 MΩ) was filtered using a Milli-Q Integral 5 water purification system (Millipore, Billerica, MA, USA).

### 3.3. Preparation of Standard and Sample Solutions for HPLC Analysis

Standard solutions were prepared using standard compounds. CGA, NA, CA, IAA, IAB, IAC, MB, MA, and DB were accurately weighed and dissolved in 50% methanol to achieve concentrations of 9.345, 1.355, 1.445, 5.050, 0.215, 0.158, 9.515, 2.635, and 3.215 mg.mL^−1^, respectively. The standard compound mixture solution was further diluted 5, 10, 15, 25, 50, and 100 times. The linear relationship for calibration was obtained by plotting the peak area against the corresponding concentration of each analyte.

Lonicerae Flos samples were heated at 60 °C for at least 30 min before use, ground into powder, and passed through a 48-mesh (300 μm) sieve. The resulting fine powder was stored away from sunlight at room temperature until analysis. Subsequently, 250 mg of fine powder from each sample was weighed and extracted in a 25 mL volumetric flask using 50% methanol under supersonic (300 W) conditions for 60 min. After cooling the solution to room temperature, methanol was added to the scale line and the solution was centrifuged at 5000 rpm for 10 min. The supernatant was then filtered through a 0.45 μm filter (Tianjin Jinteng Experiment Equipment Co., Ltd., Tianjin, China) for HPLC analysis. Each sample was prepared in triplicate.

### 3.4. HPLC Conditions

The HPLC system (Agilent, Santa Clara, CA, USA) was equipped using a G1311A quat pump, G1322A degasser, G1329A auto-sampler, G1316A column oven, and G1315B DAD detector. HPLC analysis was conducted using an Agilent 1200 system (Agilent, USA) with an RD C18 column (4.6 mm × 250 mm, 3 μm). The mobile phase comprised Solvent A (0.1% formic acid in water, *v*/*v*) and Solvent B (acetonitrile), with a gradient elution profile: 0 min, 10% B; 20 min, 20% B; 35 min, 30% B; 40 min, 35% B; 45 min, 50% B; 50 min, 50% B; and 51 min, 10% B. The flow rate of the mobile phase was 0.6 mL/min, and the column temperature was maintained at 35 °C. Detection was conducted at a wavelength of 326 nm, with an injection volume of 5 μL.

Evaporative light scattering detection was conducted using an Alltech ELSD 2000ES instrument (Champaign, IL, USA). Dry condensed air served as the carrier gas at a flow rate of 2 L/min, whereas the detector temperature was maintained at 55 °C.

### 3.5. HPLC Method Validation

Precision was evaluated by conducting six analyses of a standard solution containing nine standard compounds, and the relative standard deviation (RSD) of the peak area for each standard compound was calculated. Repeatability was assessed by analyzing six sample solutions, with variations expressed as RSD. One sample solution was analyzed at 0, 1, 2, 4, 6, 8, 16, and 24 h to investigate stability. The accuracy of the method was assessed using a recovery test. Standard compounds were weighed to approximately 100% of their original amount and added to the CQ3 sample powder. This sample, containing the standard compounds, was then extracted and analyzed using the aforementioned method.

### 3.6. Sample Process for Oxidation Resistance Assay

To assess the oxidative resistance activities of the Lonicerae Flos samples and identify the compounds contributing to these activities, 32 samples derived from the unopened type of *L. macranthoides* were selected for an oxidation resistance assay. For each sample, 0.02 g of fine powder was placed in a 15 mL centrifugation tube and 10 mL of distilled water was added. The powder was then subjected to extraction under supersonic (300 W) conditions for 30 min. Thereafter, the solution was centrifuged at 10,000 rpm for 10 min, and the supernatant was used for the analysis. Each sample was analyzed in triplicate.

### 3.7. ABTS Free Radical Scavenging Assay

The ABTS radical cation was prepared by mixing 3 mL of ABTS (7.4 mM) with 3 mL of K_2_S_2_O_8_ (2.6 mM) for 12–16 h at room temperature in the dark. The resulting mixture was diluted with distilled water until it reached an absorbance of 0.70 ± 0.10 at 734 nm. Next, 20 μL of the sample extraction was combined with 180 μL of the ABTS radical cation solution and allowed to react for 10 min in darkness at room temperature. The absorbance was then measured at 734 nm using a microplate reader (TECAN DNA Export, Morrisville, NC, USA), and the resulting value was recorded as At. To establish the baseline, 200 μL of distilled water was measured as a blank (Ab). In addition, 20 μL of distilled water, used as the sample powder and processed as described in the previous section, served as the negative control, and its final absorbance was recorded as A0. The ABTS radical clearing ratio was calculated using Formula (1): ABTS radical scavenging activity = [A0 − (At − Ab)]/A0 × 100%. (1)

### 3.8. DPPH-Free Radical Scavenging Assay

Initially, 50 μL of the sample extraction was combined with 1000 μL of freshly prepared 0.5 M DPPH radical cation solution, gently mixed, and allowed to react for 40 min in the dark at room temperature. Subsequently, the absorbance was measured at 517 nm using a microplate reader (TECAN DNA Export, USA), and the resulting value was recorded as Ax0. For the assay, 50 μL of absolute ethyl alcohol was used as the sample powder and extracted as described in the previous section, and its absorbance was recorded as A0. To eliminate the color contribution from the sample powder, the absorbance of 50 μL of the extraction mixed with 100 μL of absolute ethyl alcohol was measured and recorded as Ax. The DPPH radical clearing ratio was calculated using Formula (2): DPPH radical scavenging activity = [A0 − (Ax0 − Ax)]/A0 × 100%.(2)

### 3.9. Multivariate Analysis

OPLS-DA was conducted on the 50 Lonicerae Flos samples derived from *L. macranthoides* to characterize the components of samples from different regions and breeds. PCA, PLS regression, and heatmap analysis were conducted to elucidate the relationship between HPLC chromatograms and antioxidant activity. The data matrix used for OPLS-DA was created using the HPLC peak areas of the 50 samples. By contrast, the data matrix used for PCA, PLS regression, and heatmap analysis was created using HPLC peak areas and ABTS/DPPH scavenging activity data of 32 crude drug samples. Both data matrices were normalized using the Z-score transformation method, as described in our previous studies [46,47]. SIMCA-P (Version 14.1, Umetrics, Umea, Sweden) was used for OPLS-DA, PCA, and PLS regression, whereas Origin 2021 (Version 9.8; MicroCal, LLC, Malvern, UK) was used for heatmap analysis

## 4. Conclusions

Fifty samples of Lonicerae Flos were collected that were sourced from various regions and different breeds in Southwest China. These samples underwent quantitative analysis using HPLC for six organic acids and three saponin compounds alongside antioxidant assays using ABTS and DPPH radical scavenging experiments. All samples met the quality standards outlined in the Chinese Pharmacopoeia and exhibited high quality owing to their elevated levels of nine bioactive compounds. In particular, samples from Chongqing and Hunan displayed similar quality, those from Sichuan exhibited relatively lower quality, and samples from Guizhou, particularly the unopened and open wild types, demonstrated higher quality. However, relatively significant fluctuations in quality were observed among samples derived from the open wild-type. In addition, all samples exhibited strong radical scavenging abilities in the ABTS and DPPH assays, with the tendencies observed in the bioactive compound contents. Multivariate analysis identified CGA, IAA, IAB, and IAC as active antioxidant compounds in Lonicerae Flos. These results contribute to the quality control of Lonicerae Flos and provide a foundation for cultivating *L. macranthoides*.

## Figures and Tables

**Figure 1 molecules-29-02560-f001:**
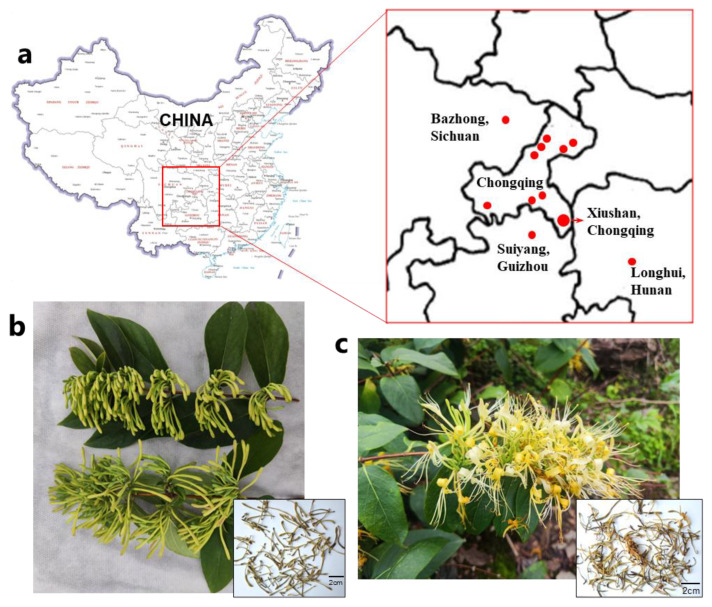
Different Lonicerae Flos-producing regions and *L. macranthoides* breeds. (**a**) Producing regions of Lonicerae Flos in this study. (**b**) Unopened-type *L. macranthoides* and the derived Lonicerae Flos. (**c**) Open wild type of *L. macranthoides* and the derived Lonicerae Flos.

**Figure 2 molecules-29-02560-f002:**
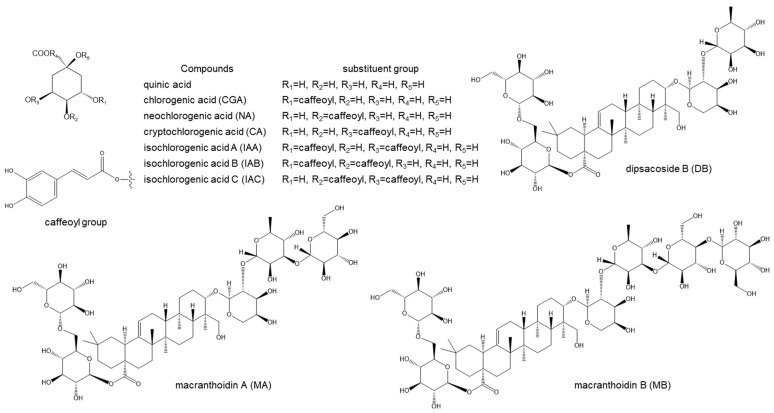
Structures of reference compounds.

**Figure 3 molecules-29-02560-f003:**
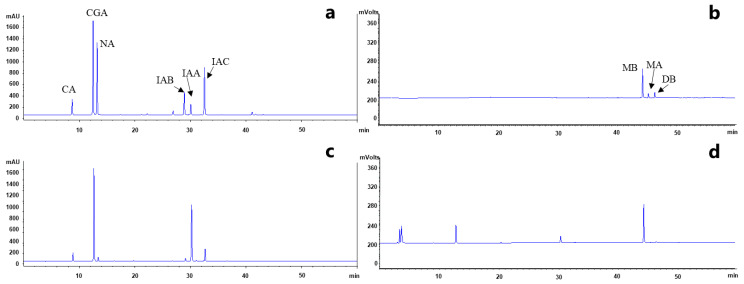
Representative HPLC chromatograms of *L. macranthoides* samples and standard compounds. CGA: chlorogenic acid; NA: neochlorogenic acid; CA: cryptochlorogenic acid; IAA: isochlorogenic acid A; IAB: iso-chlorogenic acid B; IAC: isochlorogenic acid C; MB: macranthoidin B; MA: macranthoidin A; DB: dipsacoside B. (**a**) HPLC-DAD chromatogram of six chemical standard phenolic acid compounds. (**b**) HPLC-ELSD chromatograms of three chemical standard saponin compounds. (**c**) HPLC-DAD chromatogram of sample CQ3. (**d**) HPLC-ELSD chromatogram of sample CQ3.

**Figure 4 molecules-29-02560-f004:**
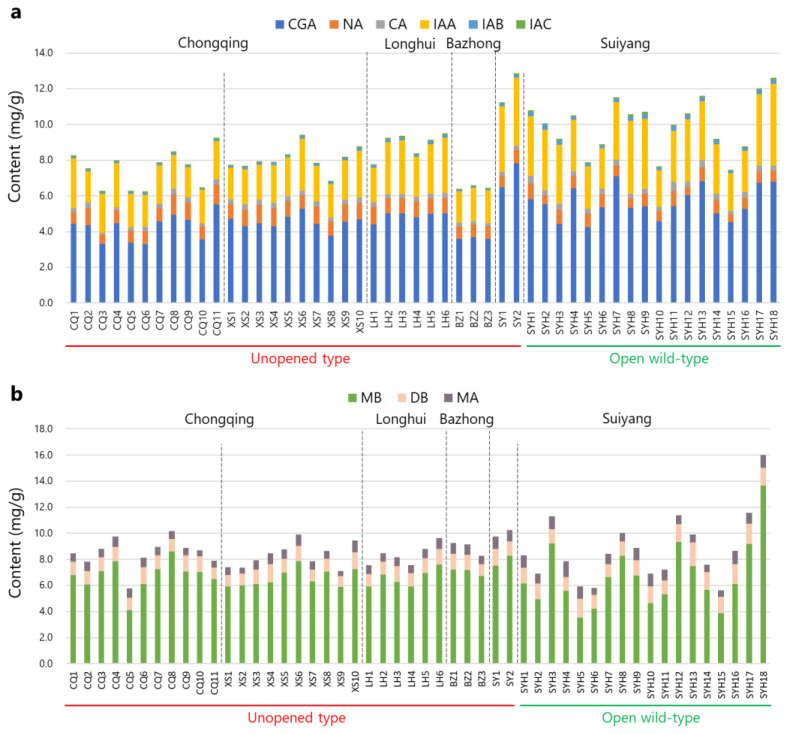
Quantitative results of *L. macranthoides*: (**a**) relative contents of six phenolic acid compounds and (**b**) three saponin compounds. The samples were distinguished by dashed lines according to different producing areas or breeds.

**Figure 5 molecules-29-02560-f005:**
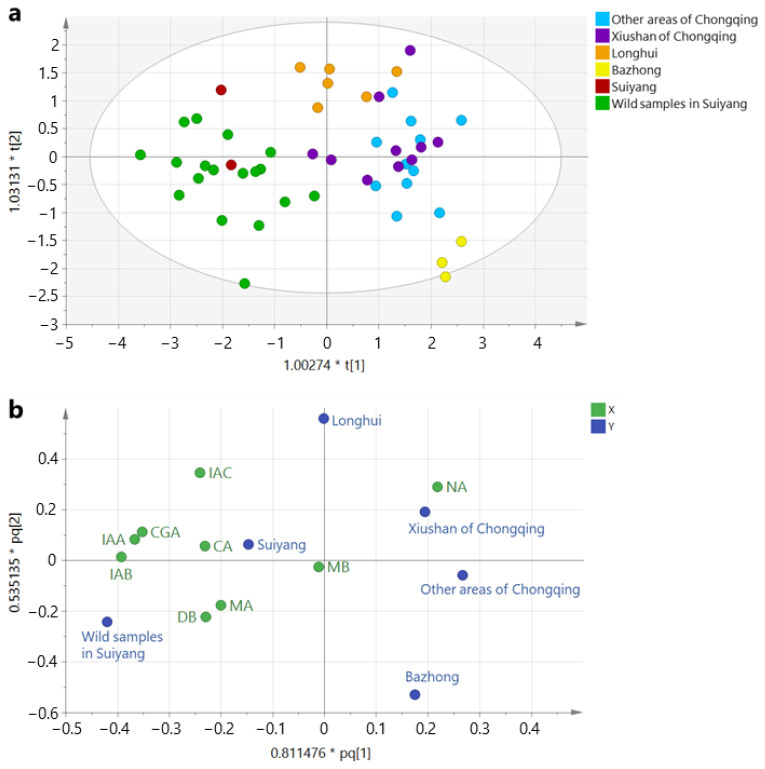
OPLS-DA results of 50 *L. macranthoides* samples: (**a**) score plot, circles with different colors indicate the samples produced in different regions of China; (**b**) loading plot, X (green) indicates compounds analyzed in this study, and Y (blue) indicates the different producing regions (response variables) of the samples.

**Figure 6 molecules-29-02560-f006:**
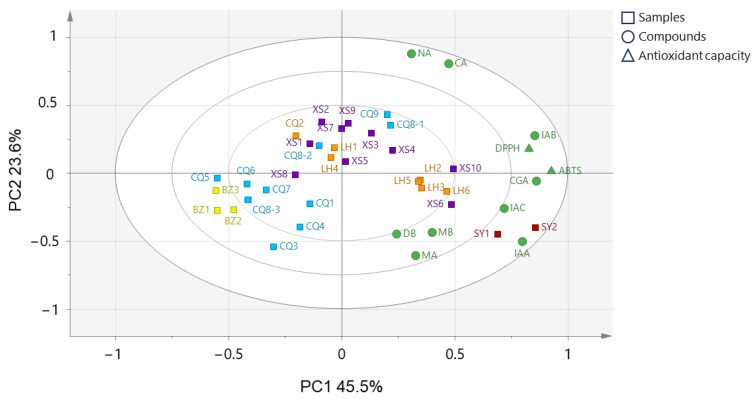
PCA results of 32 *L. macranthoides* samples using the data matrix created by HPLC quantitative data and antioxidant capacity. The square indicates each sample, the circle indicates compounds, and the triangle indicates the antioxidant capacity of the crude drug samples.

**Figure 7 molecules-29-02560-f007:**
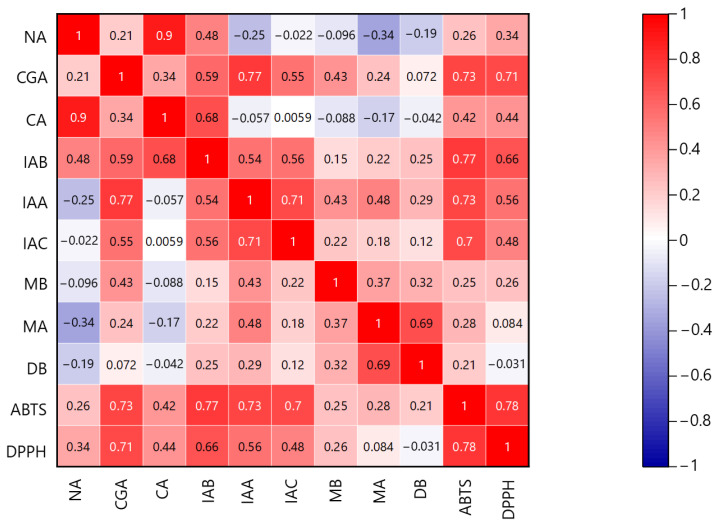
Heatmap using Pearson’s correlation analysis of the compound contents and antioxidant capacity of crude drug samples. The numerals on the graph indicate Pearson’s correlation coefficient; alphabetical abbreviations indicate compound names; and ABTS and DPPH indicate ABTS and DPPH free radical scavenging activity, respectively.

**Figure 8 molecules-29-02560-f008:**
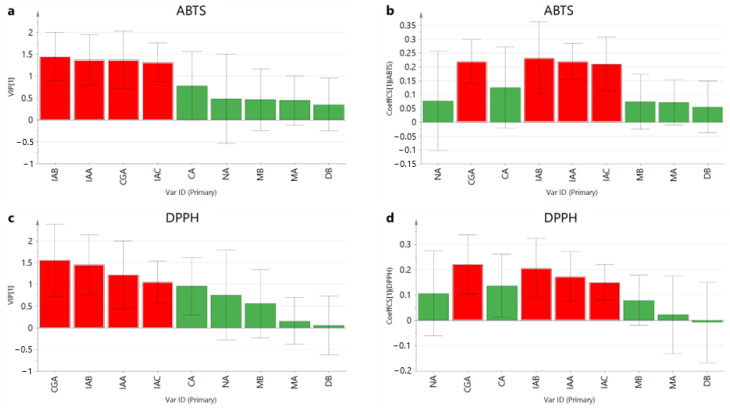
VIP score and coefficient plot of PLS regression using compounds (X variables) and antioxidant activity (Y variables). (**a**,**b**) are under the ABTS scavenging activity condition, and (**c**,**d**) are under the DPPH scavenging activity condition.

**Table 1 molecules-29-02560-t001:** Regression equations, correlation coefficients, linearity ranges, precisions, repeatability, stability, and recovery for nine analytes.

Component	Calibration Curve Equation	r	Linear Range (mg/mL)	Precision (RSD%) *n* = 6	Repeatability (RSD%) *n* = 6	Stability (RSD%) *n* = 8	Recovery (%) *n* = 6	RSD%
CGA	y = 23722x + 494.29	0.9993	0.098–0.935	1.57	0.42	0.36	100.50	1.45
NA	y = 23991x + 85.449	0.9999	0.014–0.271	1.95	0.24	0.36	100.67	1.13
CA	y = 18329x + 58.61	0.9999	0.015–0.289	1.60	2.60	1.40	99.00	1.91
IAA	y = 27927x + 240.89	0.9999	0.053–0.505	1.90	1.40	0.34	93.99	1.55
IAB	y = 22053x + 12.829	0.9997	0.002–0.043	1.48	1.46	1.66	99.89	1.22
IAC	y = 23732x − 70.103	0.9996	0.002–0.032	1.73	1.81	1.39	99.89	1.61
MB	y = 732336x − 79386	0.9983	0.010–1.903	1.14	0.13	1.14	100.41	1.53
MA	y = 503752x − 9198.3	0.9926	0.028–0.527	0.79	2.13	1.09	99.56	1.47
DB	y = 482044x − 25891	0.9978	0.033–0.625	0.97	0.70	1.63	100.13	1.59

CGA: chlorogenic acid; NA: neochlorogenic acid; CA: cryptochlorogenic acid; IAA: isochlorogenic acid A; IAB: iso-chlorogenic acid B; IAC: isochlorogenic acid C; MB: macranthoidin B; MA: macranthoidin A; DB: dipsacoside B; r: correlation coefficient; RSD: relative standard deviation.

**Table 2 molecules-29-02560-t002:** The contents of nine compounds in samples.

Producing Area	Voucher No.	CGA	NA	CA	IAA	IAB	IAC	SUM of Six Phenolic Acids	MB	DB	MA	SUM of MB&DB	SUM of Three Saponins
Chongqing	CQ1	4.426	0.667	0.200	2.804	0.092	0.082	8.270	6.797	1.004	0.651	7.802	8.452
	CQ2	4.357	0.977	0.300	1.738	0.096	0.075	7.542	6.075	1.020	0.736	7.094	7.830
	CQ3	3.326	0.496	0.117	2.167	0.056	0.116	6.278	7.103	1.044	0.660	8.147	8.807
	CQ4	4.455	0.748	0.171	2.463	0.069	0.097	8.003	7.850	1.117	0.795	8.967	9.763
	CQ5	3.359	0.687	0.185	1.876	0.086	0.087	6.281	4.102	0.954	0.704	5.056	5.761
	CQ6	3.296	0.721	0.245	1.810	0.107	0.079	6.257	6.120	1.288	0.726	7.407	8.133
	CQ7	4.576	0.763	0.224	2.140	0.086	0.082	7.870	7.234	1.064	0.651	8.298	8.949
	CQ8	4.947	1.144	0.314	1.886	0.125	0.089	8.505	8.622	0.942	0.592	9.564	10.156
	CQ9	4.658	0.990	0.274	1.671	0.102	0.087	7.782	7.057	1.263	0.560	8.320	8.880
	CQ10	3.569	0.718	0.186	1.858	0.078	0.077	6.486	7.039	1.214	0.448	8.253	8.701
	CQ11	5.539	1.077	0.315	2.124	0.102	0.087	9.244	6.479	0.887	0.540	7.366	7.906
	XS1	4.711	0.837	0.268	1.746	0.098	0.080	7.739	5.923	0.878	0.610	6.801	7.411
	XS2	4.287	0.938	0.291	1.979	0.112	0.074	7.682	6.014	0.893	0.470	6.906	7.376
	XS3	4.465	1.026	0.304	1.934	0.122	0.099	7.950	6.111	1.102	0.714	7.213	7.926
	XS4	4.292	1.026	0.301	2.095	0.126	0.082	7.923	6.220	1.413	0.849	7.632	8.481
	XS5	4.829	0.908	0.254	2.161	0.102	0.081	8.335	6.995	1.064	0.697	8.059	8.756
	XS6	5.281	0.748	0.247	2.911	0.145	0.105	9.437	7.850	1.186	0.865	9.036	9.900
	XS7	4.431	0.955	0.298	1.992	0.112	0.075	7.864	6.291	0.939	0.612	7.230	7.842
	XS8	3.770	0.801	0.209	1.865	0.102	0.092	6.838	7.074	0.974	0.610	8.048	8.658
	XS9	4.549	0.983	0.261	2.191	0.110	0.099	8.193	5.899	0.836	0.350	6.735	7.086
	XS10	4.677	0.917	0.297	2.622	0.156	0.098	8.768	7.240	1.311	0.895	8.551	9.446
Hunan	LH1	4.423	0.973	0.242	1.935	0.108	0.103	7.783	5.930	0.950	0.688	6.880	7.567
	LH2	5.033	0.827	0.243	2.911	0.132	0.115	9.261	6.818	1.006	0.631	7.824	8.456
	LH3	5.025	0.828	0.241	3.019	0.126	0.119	9.358	6.275	1.201	0.690	7.477	8.167
	LH4	4.805	0.896	0.250	2.237	0.108	0.097	8.393	5.923	1.019	0.601	6.942	7.543
	LH5	4.997	0.867	0.257	2.778	0.131	0.115	9.145	6.942	1.144	0.730	8.086	8.816
	LH6	5.020	0.885	0.273	3.071	0.144	0.114	9.508	7.586	1.200	0.868	8.786	9.654
Sichuan	BZ1	3.598	0.708	0.190	1.772	0.065	0.063	6.395	7.198	1.241	0.833	8.439	9.272
	BZ2	3.687	0.709	0.186	1.884	0.067	0.065	6.598	7.160	1.173	0.821	8.332	9.153
	BZ3	3.608	0.681	0.181	1.848	0.064	0.061	6.442	6.712	0.914	0.652	7.626	8.278
Guizhou	SY1	6.483	0.654	0.223	3.660	0.130	0.105	11.255	7.503	1.319	0.921	8.822	9.743
	SY2	7.839	0.753	0.228	3.797	0.127	0.120	12.864	8.264	1.108	0.871	9.371	10.243
	SYH1	5.808	0.906	0.416	3.332	0.234	0.109	10.805	6.158	1.206	0.946	7.364	8.310
	SYH2	5.533	0.487	0.287	3.404	0.264	0.105	10.080	4.951	1.216	0.747	6.167	6.914
	SYH3	4.447	0.770	0.337	3.297	0.232	0.103	9.185	9.215	1.110	0.984	10.325	11.308
	SYH4	6.425	0.691	0.274	2.869	0.170	0.095	10.525	5.583	1.069	1.202	6.652	7.854
	SYH5	4.241	0.790	0.254	2.375	0.139	0.088	7.887	3.551	1.438	0.939	4.989	5.928
	SYH6	5.358	0.725	0.305	2.273	0.156	0.079	8.896	4.220	1.057	0.520	5.277	5.797
	SYH7	7.095	0.630	0.289	3.239	0.182	0.093	11.528	6.647	0.974	0.816	7.621	8.438
	SYH8	5.332	0.532	0.262	4.088	0.248	0.118	10.581	8.282	1.103	0.624	9.385	10.008
	SYH9	5.404	0.713	0.290	3.914	0.263	0.118	10.700	6.757	1.192	0.940	7.949	8.889
	SYH10	4.587	0.550	0.252	2.037	0.141	0.083	7.650	4.619	1.319	0.957	5.939	6.896
	SYH11	5.408	0.879	0.490	2.880	0.249	0.085	9.991	5.331	1.050	0.827	6.381	7.207
	SYH12	6.033	0.450	0.308	3.500	0.233	0.105	10.629	9.345	1.343	0.698	10.689	11.387
	SYH13	6.831	0.778	0.393	3.294	0.203	0.108	11.608	7.492	1.789	0.635	9.281	9.915
	SYH14	5.038	0.728	0.363	2.770	0.217	0.090	9.206	5.671	1.361	0.550	7.032	7.583
	SYH15	4.507	0.464	0.194	2.100	0.122	0.076	7.464	3.888	1.230	0.514	5.118	5.632
	SYH16	5.287	0.606	0.295	2.339	0.166	0.074	8.766	6.119	1.529	1.010	7.648	8.658
	SYH17	6.738	0.627	0.329	3.993	0.229	0.111	12.027	9.168	1.584	0.813	10.751	11.565
	SYH18	6.793	0.607	0.285	4.582	0.219	0.139	12.625	13.652	1.348	1.016	15.000	16.015

CGA: chlorogenic acid; NA: neochlorogenic acid; CA: cryptochlorogenic acid; IAA: isochlorogenic acid A; IAB: iso-chlorogenic acid B; IAC: isochlorogenic acid C; MB: macranthoidin B; MA: macranthoidin A; DB: dipsacoside B.

**Table 3 molecules-29-02560-t003:** The clearing ratio of water extraction of unopened *L. macranthoides* samples.

Voucher No.	ABTS		DPPH	
	Clearing Ratio %	SD	Clearing Ratio %	SD
CQ1	77.89	0.03	58.24	0.06
CQ2	77.07	0.02	55.50	0.02
CQ3	81.80	0.02	58.09	0.05
CQ4	77.14	0.02	53.85	0.05
CQ5	71.89	0.02	52.65	0.07
CQ6	72.07	0.02	48.45	0.06
CQ7	69.61	0.03	51.24	0.07
CQ8	81.74	0.03	62.59	0.06
CQ9	77.01	0.02	57.19	0.06
CQ10	72.46	0.02	55.83	0.11
CQ11	82.62	0.01	68.18	0.02
XS1	80.80	0.02	61.12	0.08
XS2	81.75	0.02	61.34	0.05
XS3	82.34	0.05	62.24	0.04
XS4	90.76	0.02	62.26	0.05
XS5	77.01	0.04	65.95	0.03
XS6	87.44	0.01	67.07	0.05
XS7	81.83	0.01	64.25	0.03
XS8	73.47	0.02	62.16	0.02
XS9	81.62	0.01	65.69	0.04
XS10	86.45	0.01	69.66	0.03
LH1	79.03	0.02	61.72	0.04
LH2	88.38	0.01	61.69	0.03
LH3	85.93	0.01	63.89	0.04
LH4	79.21	0.03	56.56	0.05
LH5	85.85	0.02	61.76	0.02
LH6	87.60	0.02	59.67	0.04
BZ1	68.39	0.01	52.86	0.04
BZ2	72.76	0.01	53.68	0.05
BZ3	71.57	0.03	56.06	0.11
SY1	92.12	0.01	71.37	0.05
SY2	92.52	0.01	69.86	0.06

**Table 4 molecules-29-02560-t004:** Lonicerae Flos (derived from *L. macranthoides*) samples localities, voucher number, breeds, and harvest date.

Producing Area		Voucher No.	*L. macranthoides* Breeds	Collection Date
Chongqing	Zhong county	CQ1	unopened type	2022.6
	Jiangjin district	CQ2	unopened type	2022.6
	Kaizhou district	CQ3	unopened type	2022.7
	Wanzhou district	CQ4	unopened type	2022.7
	Pengshui county	CQ5	unopened type	2022.7
	Kaizhou district	CQ6	unopened type	2022.7
	Wulong district	CQ7	unopened type	2022.7
	Fengjie county	CQ8	unopened type	2022.7
	Fengjie county	CQ9	unopened type	2022.7
	Fengjie county	CQ10	unopened type	2022.7
	Yunyang county	CQ11	unopened type	2022.7
	Liangshui village, Longfengba town, Xiushan county	XS1	unopened type	2022
	Shiban village, Rongxi town, Xiushan county	XS2	unopened type	2022
	Shuiyuan village, Longchi town, Xiushan county	XS3	unopened type	2022
	Guiluo village, Zhongping town, Xiushan county	XS4	unopened type	2022
	Pingyang village, Qingxichang, Xiushan county	XS5	unopened type	2022
	Malu, Zhongling town, Xiushan county	XS6	unopened type	2022
	Rongxi town, Xiushan county	XS7	unopened type	2022
	Pingjian, Pingkai, Xiushan county	XS8	unopened type	2022
	Xinqiao village, Cenxi town, Xiushan county	XS9	unopened type	2022
	Bamang village, Aikou town, Xiushan county	XS10	unopened type	2022
Hunan	Longhui county, Shaoyang city	LH1	unopened type	2022
	Longhui county, Shaoyang city	LH2	unopened type	2022
	Longhui county, Shaoyang city	LH3	unopened type	2022
	Longhui county, Shaoyang city	LH4	unopened type	2022
	Longhui county, Shaoyang city	LH5	unopened type	2022
	Longhui county, Shaoyang city	LH6	unopened type	2022
Sichuan	Bazhong city	BZ1	unopened type	2022.8
	Bazhong city	BZ2	unopened type	2022.8
	Bazhong city	BZ3	unopened type	2022.8
Guizhou	Suiyang county, Zunyi city	SY1	unopened type	2022.9
	Suiyang county, Zunyi city	SY2	unopened type	2022.9
	Suiyang county, Zunyi city	SYH1	open wild type	2022.7
	Suiyang county, Zunyi city	SYH2	open wild type	2022.7
	Suiyang county, Zunyi city	SYH3	open wild type	2022.7
	Suiyang county, Zunyi city	SYH4	open wild type	2022.7
	Suiyang county, Zunyi city	SYH5	open wild type	2022.7
	Suiyang county, Zunyi city	SYH6	open wild type	2022.7
	Suiyang county, Zunyi city	SYH7	open wild type	2022.7
	Suiyang county, Zunyi city	SYH8	open wild type	2022.7
	Suiyang county, Zunyi city	SYH9	open wild type	2022.7
	Suiyang county, Zunyi city	SYH10	open wild type	2022.7
	Suiyang county, Zunyi city	SYH11	open wild type	2022.7
	Suiyang county, Zunyi city	SYH12	open wild type	2022.7
	Suiyang county, Zunyi city	SYH13	open wild type	2022.7
	Suiyang county, Zunyi city	SYH14	open wild type	2022.7
	Suiyang county, Zunyi city	SYH15	open wild type	2022.7
	Suiyang county, Zunyi city	SYH16	open wild type	2022.7
	Suiyang county, Zunyi city	SYH17	open wild type	2022.7
	Suiyang county, Zunyi city	SYH18	open wild type	2022.7

## Data Availability

The data presented in this study are available upon request from the corresponding author.

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
