# Peer review of "Quality Evaluation of Lonicerae Flos Produced in Southwest China Based on HPLC Analysis and Antioxidant Activity"

_molecules, 2024, doi:10.3390/molecules29112560_

Round 1

Reviewer 1 Report

Comments and Suggestions for Authors

This paper provides valuable insights into the antioxidant activity of Lonicerae Flos samples derived from L. macranthoides. However, it could be accepted only after the following major points are addressed carefully by the authors.

Introduction: Add more information on chemical and genomic diversity of Lonicera species mentioning the paper DOI: 10.1016/j.phytochem.2018.07.012

Better state the importance of antioxidant compounds from plants in traditional medicine. Recent examples from the literature of antioxidant properties from vegetal sources, including plants from Caucasus should be added. Cite at least the paper DOI: 10.2174/0109298673262575231127034952

lines 51-52: no need of capital letters for ' Hederin-type, Olean- 51

ane-type, Ursane-type, Lupane-type, Fernane-1-type, and Fernane-2-type'

Tab 2: 

  • Consider presenting the data in Table 2 more concisely by grouping the compounds contents according to their producing regions

Figure 1: provide a better quality picture as the numbers are not readable

Provide a new Figure or Scheme reporting the chemical structures of the compounds whose abbreviations are reported in Figure 1

Figure 2: It is very difficult to examine such Figure. Can you divide into two parts highlighting better the contents? Moreover, I see an overlapping between two pictures which makes the comprehension even harder. Some writings are covered in this way by the green bars. Moreover, should not you provide error bars for the contents reported?

Figure 3: the legend should follow the figure in my opinion

Figure 4: hexagons and cyrcles can not be distinguished easily. Provide a better figure.

Conclusions: emphasize the significance of the results obtained for understanding the antioxidant activity of the studied samples in the context of the conclusive section. It must be improved significantly in my opinion.

Comments on the Quality of English Language

English is generally fine.

Author Response

Dear reviewer:

Thank you for your precious time on reviewing our manuscript. We have revised the manuscript according to your comments. Your suggestions helped me a lot with this article and my research in the future.

Reviewer 2 Report

Comments and Suggestions for Authors

Minor remarks

Please, provide a blank space between quantity and unit only except in the case of percentage.

Some sentences are written using the first-person plural. The scientific paper is necessary to write using the third-person singular. Having this in mind, all text should be retyped according to this recommendation.

Trans should be depicted in italics.

Watt unit should be presented using the capital letter. Use also capital letter when present the unit of milliliter (mL).

The equation should be numbered in the manuscript.

All minor remarks are depicted in the document file.

Major remarks

In the Introduction, I noticed a lumping of references (for instance, [10-15]) without any deep discussion. My recommendation is to discuss the reference separately.

The main contribution and significance of this research should be better depicted. In this form, the chemical analysis of various plant species is only presented. The manuscript with this conception is not interesting.

Maybe, the manuscript should focus more on the advantages and possibility of further using plant species based on their chemical composition.

Comments on the Quality of English Language

English is acceptable and does not require serious modification.

Author Response

(The authors gave the same response as above.)

Reviewer 3 Report

Comments and Suggestions for Authors

The article is written in a clear and efficient english. Additionally, the investigation is very interesting, however some improvments must be applied to enhance the quality of this article:

1. correct the "antioxidant activity" on the title.

2. correct line 87 on the introduction.

3. put the all the tables and figures right after their mention

4. add the abreviations meaning down the tables.

5. in figure 1 : the labels (a,b,c,d) should not interfer with the graph axis.

6. Correct table 2 title.

7. correct figure 2 (image shift).

8. Figure 3 title must be put down of the figure

9. Table 3 (which concentrations are used on the ABTS and DPPH tests ?)

10. make the figures 3, 5, 6 and 7 bigger.

11. correct table 4 title.

12. mention figure 7 on the text.

13. in both DPPH and ABTS methods mention what is bleaching ratio % is it representing you result explain what it means, and mention that you tested each sample 3 times.

14. I remark that you calculated the STDEV of the samples which test u conducted which p-value you used? and which software ? u can add a subtitle "statistical analysis".

15. In the conclusions end, give perspective about how your results can be exploited on the future, try to valorize your findings more.

Author Response

(The authors gave the same response as above.)

Round 2

Reviewer 1 Report

Comments and Suggestions for Authors

The paper can be accepted in the current form 

Comments on the Quality of English Language

English is generally fine